# Field Trials to Assess the Growth, Survival, and Stomatal Densities of Five Mexican Pine Species and Their Hybrids under Common Plantation Conditions

Ricardo Silas Sánchez-Hernández [1], Carmen Zulema Quiñones-Pérez [2], José Ciro Hernández-Díaz [3], José Ángel Prieto-Ruíz [4] and Christian Wehenkel [3,*]

1    Maestría Institucional en Ciencias Agropecuarias y Forestales (MICAF), Universidad Juárez del Estado de Durango (UJED), Durango 34120, Mexico
2    Tecnológico Nacional de México Campus Valle del Guadiana (TecNM-ITVG), Villa Montemorelos, Durango 34371, Mexico
3    Instituto de Silvicultura e Industria de la Madera (ISIMA), Universidad Juárez del Estado de Durango (UJED), Durango 34120, Mexico
4    Facultad de Ciencias Forestales y Ambientales (FCFA), Universidad Juárez del Estado de Durango (UJED), Durango 34120, Mexico
*    Correspondence: wehenkel@ujed.mx

**Abstract:** Understanding hybridization is important for practical reasons, as the presence of hybrid trees in seed stands can influence the success of natural regeneration and reforestation. Hybridization creates new gene combinations, which can promote or enhance adaptation to new or changing environments. In the present research, we aimed, for the first time, to evaluate and compare the growth and survival of 541 putative hybrid seedlings and 455 seedlings of the pure parental trees of *Pinus arizonica*, *P. durangensis*, *P. engelmannii*, *P. leiophylla,* and *P. teocote*, in two reciprocal trials of duration 27 months in the Sierra Madre Occidental (SMO), Durango, Mexico. We also examined the possible correlation between needle stomatal density and seedling growth and survival. The overall analysis of the data showed that the mean height to the apical bud was significantly higher ($p = 0.01$) in the hybrids than in the pure trees. Considering both trials, the survival rate of *P. arizonica* ($p = 0.002$) and *P. durangensis* ($p = 0.01$) hybrids was significantly higher than that of the pure trees. The growth parameters were significantly correlated with the mean stomatal density ($p < 0.01$). Stomatal density and survival at the seed stand level were significantly and positively correlated in the hybrids, but not in the pure trees. In summary, *Pinus* hybrids generally exhibited the same ability as the pure species (or sometimes a greater ability) to withstand weather conditions, survive, and grow effectively in both growth trials. The systematic use of natural pine hybrids in Mexico could therefore be considered a possible option for sustainable management and as a component of adaptive silviculture.

**Keywords:** natural hybridization; *Pinus arizonica*; *Pinus durangensis*; *Pinus engelmannii*; *Pinus leiophylla*; *Pinus teocote*

## 1. Introduction

Natural hybridization is a common phenomenon in plants [1], as more than 25% of plants hybridize naturally [2]. Interspecific gene transfer occurs during hybridization, which may introduce more different genetic material than that generated directly by mutations [3]. Understanding hybridization is important for practical reasons, as the presence of hybrids in seed stands can influence the success of natural regeneration and reforestation [4,5]. In the same generation of hybrids, viability, fertility, and vigor can vary widely across individuals, with some of them having the same values, lower values, or even higher values than their parents [4]. Different cases of natural hybridization of Mexican pine species have been observed [6].

Hernández-Velasco et al. [7] detected significant differences in the survival, diameter, and height between seedlings of the pure parental trees and putative hybrid seedlings of five very important timber species in Mexico, i.e., *Pinus arizonica*, *P. durangensis*, *P. engelmannii*, *P. leiophylla,* and *P. teocote* [8], which were grown for 15 months in nursery conditions. The seed of the species was collected from seed stands in the municipalities of Tepehuanes, Otáez, and Santiago Papasquiaro, in the state of Durango, Mexico. Because the controlled conditions in nurseries are completely different from those in natural field environments, the same authors [7] recommended further studies to determine the performance of each hybrid in field conditions, particularly in regions where slower growing parental trees are found, as well as in extreme environments. This is because hybridization creates new gene combinations that can promote or enhance adaptation to new or changing environments [9].

Provenance trials are useful for detecting associations between genetic, geographic, and climatic factors [10]. However, these trials are time-consuming and usually only allow for the measurement of phenotypic differences between individuals and populations under common conditions. Only reciprocal trials allow for the contribution of phenotypic plasticity and the interactions between genotype and environment to be revealed [11,12].

Some forest plant species have wide distribution ranges, as they possess adaptive strategies that allow them to survive and grow in ecologically different areas [13]. One of these strategies is the morphological alteration of leaves, as a consequence of stress-related effects [14]. The stomata are microscopic structures present on the surface of leaves. In the case of conifers, the stomata have a protective function as they surround a central pore and limit access to mesophyll cells. Environmental factors such as light intensity, atmospheric $CO_2$ concentration, and internal control systems regulate the development of the stomata [15]. Plants can alter the opening of the stomatal pores, moderating gas exchange between the leaf interior and the atmosphere [16]. The morphology and distribution of stomata vary in response to environmental changes and are primarily directed by genetic traits and phenotypic plasticity, representing long-term adaptations of plant species [15,17]. Characteristics of the stomata, such as size, density, and responsiveness to environmental factors are key components influencing plant growth [18].

The present research aimed, for the first time, to evaluate and compare the growth and survival of putative hybrid seedlings and seedlings of the pure parental species of *Pinus arizonica*, *P. durangensis*, *P. engelmannii*, *P. leiophylla,* and *P. teocote*, in two reciprocal trials in the Sierra Madre Occidental (SMO), in the state of Durango, Mexico. The study also aimed to examine the possible correlation between the stomatal density of the needles and the growth and survival of the seedlings. The growth of both types of seedlings may not be statistically different [19]; however, there may be significant differences between the growth of putative hybrid seedlings and seedlings of the pure parental trees in the field in terms of either hybrid vigor [20] or hybrid depression [5]. In addition, stomatal density may also influence pine seedling growth and survival [17].

## 2. Materials and Methods

### 2.1. Study Site

Two field trials (1 hectare each) were established in July 2018 to compare the growth of putative hybrid seedlings and seedlings of the pure parental trees in the field (referred to as hybrid seedlings and pure seedlings). The planting distance between the seedlings was 2 × 2 m. Regardless of the species and the type of plant (hybrid or pure), each plant was randomly included in both trials. The trial areas were cleared of tree vegetation and protected by a 1.8 m high wire fence, before the seedlings were planted. The trials included a total of 2552 seedlings, which were produced, evaluated, and classified either as hybrids (1297, mostly consisting of backcrosses, with smaller numbers of $F_2$ and subsequent hybrid generations, but no $F_1$ hybrids) or pure species (1255). The seedlings were first grown together for 15 months in the nursery [7] and then for 27 months in the field.

The first trial was established in the Ciénega de Salpica el Agua ejido, in the area known as La Mesa Alta, at an elevation of 2710 m (25.06 N, −105.77 W). The other trial was

established in the Laguna de La Chaparra ejido, in the area known as La Mesa Seca, at an elevation of 2610 m (25.12 N, −105.70 W). Both sites are located within the municipality of Santiago Papasquiaro, state of Durango, Mexico (Figure 1).

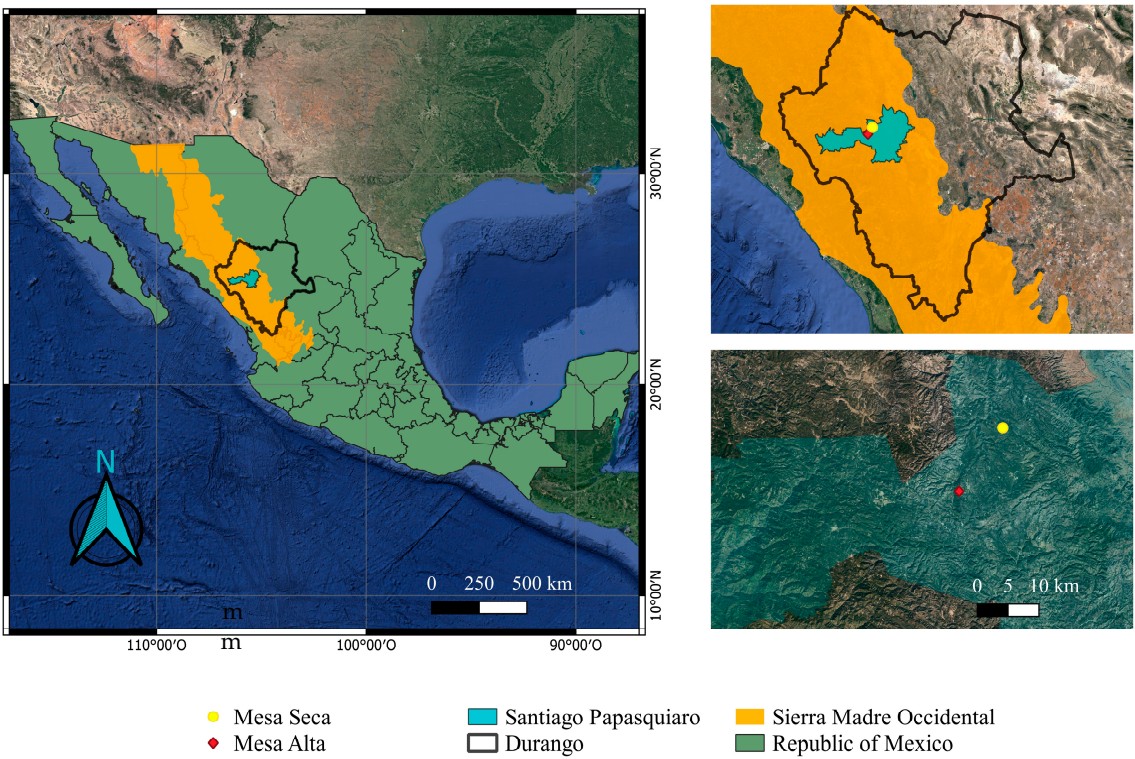

| ● Mesa Seca | ▢ Santiago Papasquiaro | ▢ Sierra Madre Occidental |
| ◆ Mesa Alta | ▢ Durango | ▢ Republic of Mexico |

**Figure 1.** Location of the two study sites where the seedlings of interest were planted and are growing.

Both sites are composed of pine–oak forests. The topography of the area consists of high and low mountain ranges. During the period 1961–1990, the mean annual temperature was 10.5 °C in Mesa Seca and 9.6 °C in Mesa Alta; the mean annual precipitation was 803 mm in Mesa Seca and 903 mm in Mesa Alta. The soil characteristics differ little between the two sites (Table 1). The presence of *Pappogeomys castanops* Baird, a rodent that consumes plants or parts of plants [21], was detected in both sites.

**Table 1.** Soil characteristics in both reciprocal trials in the municipality of Santiago Papasquiaro, state of Durango, Mexico.

| Characteristic | Mesa Alta | Mesa Seca |
|---|---|---|
| Textural class | Sandy clay loam | Loam |
| Organic matter (OM, %) | 4.64 High | 1.94 Median |
| Nitrogen (N-NO$_3$, kg/ha) | 12.32 | 7.39 |
| Phosphorus (ppm) | 9.66 | 7.39 |
| Potassium (ppm) | 220 | 116 |
| Magnesium (ppm) | 198 | 114 |
| Zinc (ppm) | 2.06 | 4.12 |
| pH 1:2 water | 5.88 | 5.25 |
| CaCO$_3$ (ppm) | 1698 | 774 |
| CEC (meq/100 g) | 10.99 | 5.35 |

CEC = cation exchange capacity.

In October 2020, seedling survival was calculated for each pure species and hybrid and per trial, as a percentage of the total number of individuals planted in both trials in July 2018 (Table 2).

**Table 2.** Descriptive statistics of the basal diameter (mm) of the surviving seedlings of the five pure *Pinus* species and their hybrids, in each separate trial and both trials together, after 27 months in the field.

| Species | Mesa Alta | | | | Mesa Seca | | | | Both Trials Together | | | |
|---|---|---|---|---|---|---|---|---|---|---|---|---|
| | N | Median | Mean | Sd | N | Median | Mean | Sd | N | Median | Mean | Sd |
| PA-H | 5 | 13.1 ab | 13.1 | 0.8 | 5 | 21.5 ab | 21.5 | 5.2 | 10 | 13.6 ab | 15.9 | 4.9 |
| PA-P | 3 | 9.6 b | 9.6 | 0.6 | 4 | 11.0 b | 11.0 | 1.4 | 7 | 10.3 ab | 10.3 | 1.0 |
| PD-H | 18 | 10.6 b | 11.4 | 3.6 | 58 | 14.8 b | 15.1 | 5.5 | 76 | 13.2 b | 14.2 | 5.3 |
| PD-P | 14 | 11.6 b | 11.5 | 2.7 | 20 | 14.6 b | 15.3 | 5.2 | 34 | 12.6 ab | 13.7 | 4.7 |
| PE-H | 56 | 19.9 a | 19.4 | 9.0 | 211 | 24.4 a | 24.6 | 7.4 | 267 | 23.6 a | 23.5 | 8.0 |
| PE-P | 78 | 20.1 a | 20.2 | 7.8 | 218 | 23.3 a | 23.6 | 7.1 | 296 | 22.5 a | 22.6 | 7.4 |
| PL-H | 7 | 17.3 ab | 19.3 | 5.9 | 18 | 20.3 ab | 20.1 | 6.7 | 25 | 20.0 b | 19.9 | 6.4 |
| PL-P | 4 | 22.1 a | 22.5 | 5.8 | 13 | 14.6 b | 17.3 | 6.9 | 17 | 18.7 b | 18.5 | 6.9 |
| PT-H | 49 | 16.2 ab | 15.9 | 6.4 | 114 | 16.7 b | 16.9 | 6.3 | 163 | 16.6 b | 16.6 | 6.3 |
| PT-P | 24 | 15.3 ab | 15.9 | 5.4 | 77 | 15.3 b | 16.8 | 5.9 | 101 | 15.3 b | 16.6 | 5.7 |
| H | 135 | 15.4 a | 15.8 | 5.1 | 406 | 19.5 a | 19.6 | 6.2 | 541 | 17.4 a | 18.0 | 6.2 |
| P | 123 | 15.7 a | 15.9 | 4.5 | 332 | 15.8 a | 16.8 | 5.3 | 455 | 15.9 a | 16.3 | 5.2 |

Sd = standard deviation, *N* = number of *Pinus* seedlings, PA-P = *Pinus arizonica*, PD-P = *P. durangensis*, PE-P = *P. engelmannii*, PL-P = *P. leiophylla* and PT-P = *P. teocote*. PA-H = hybrids of *Pinus arizonica* × *P. durangensis* genetically more similar to *P. arizonica*; PD-H = hybrids of *P. durangensis* × *P. arizonica* genetically more similar to *P. durangensis* and *P. durangensis* × *P. engelmannii* genetically more similar to *P. durangensis* PE-H = hybrids of *P. engelmannii* × *P. arizonica* genetically more similar to *P. engelmannii*; PL-H = hybrids of *P. leiophylla* × *P. teocote* genetically more similar to *P. leiophylla*; PT-H = *P. leiophylla* × *P. teocote* genetically more similar to *P. teocote*; different letters indicate significant differences (α = 0.025).

## 2.2. Evaluation of Differences in the Development of Hybrid/Pure Individuals in the Field

In total, 541 hybrid seedlings and 455 pure seedlings of the five species under study were analyzed (258 from Mesa Alta and 738 from Mesa Seca) (Table 2). The basal diameter (at plant collar) was measured using a digital Vernier scale, with a resolution of tenths of a millimeter (AVEDISTANT, LCD6); plant height at the apical growth bud and maximum needle height (from the base to the top of the needles) were measured using a flexometer, with a resolution of millimeters (Uline Accu-Lock, H-1766) (Tables 2–4). Needle length in young seedlings is considered a good indicator of future growth [22]. According to Squillance and Silen [23], pine needle length is positively correlated with height growth and thus with productivity.

**Table 3.** Descriptive statistics of height to the apical bud (cm) of *Pinus* seedlings per species and their hybrids, in each trial and both trials together, after growing for 27 months in the field.

| Species | Mesa Alta | | | | Mesa Seca | | | | Both Trials Together | | | |
|---|---|---|---|---|---|---|---|---|---|---|---|---|
| | N | Median | Mean | Sd | N | Median | Mean | Sd | N | Median | Mean | Sd |
| PA-H | 5 | 22.2 ab | 22.2 | 5.9 | 5 | 16.0 ab | 16.0 | 5.1 | 10 | 18.0 ab | 20.1 | 5.5 |
| PA-P | 3 | 30.0 a | 30.0 | 7.6 | 4 | 18.3 ab | 18.3 | 8.8 | 7 | 24.2 ab | 24.2 | 8.3 |
| PD-H | 18 | 33.2 a | 34.6 | 14.1 | 58 | 31.3 a | 32.7 | 12.2 | 76 | 31.5 a | 33.2 | 12.6 |
| PD-P | 14 | 31.3 a | 31.4 | 10.5 | 20 | 35.6 a | 36.9 | 14.7 | 34 | 32.8 a | 34.6 | 13.2 |
| PE-H | 56 | 14.2 b | 16.5 | 12.7 | 211 | 15.9 b | 16.6 | 6.7 | 267 | 15.2 b | 16.7 | 8.4 |
| PE-P | 78 | 14.2 b | 15.1 | 6.2 | 218 | 15.8 b | 17.6 | 8.3 | 296 | 15.3 b | 16.9 | 7.8 |
| PL-H | 7 | 30.0 a | 30.8 | 7.4 | 18 | 34.1 a | 33.3 | 9.1 | 25 | 31.2 a | 32.6 | 8.6 |
| PL-P | 4 | 43.3 a | 40.8 | 13.3 | 13 | 33.0 a | 34.4 | 8.2 | 17 | 37.0 a | 35.9 | 9.6 |
| PT-H | 49 | 33.7 a | 35.2 | 12.6 | 114 | 30.0 a | 31.3 | 10.9 | 163 | 32.0 a | 32.5 | 11.5 |
| PT-P | 24 | 34.5 a | 36.1 | 14.7 | 77 | 34.7 a | 36.0 | 17.2 | 101 | 34.5 a | 36.0 | 16.6 |
| H | 135 | 26.7 a | 27.9 | 10.5 | 406 | 25.5 a | 26.0 | 8.8 | 541 | 22.0 a | 24.6 | 12.9 |
| P | 123 | 30.7 b | 30.7 | 10.5 | 332 | 27.5 a | 28.6 | 11.4 | 455 | 19.5 b | 23.3 | 14.1 |

Sd = standard deviation, *N* = number of *Pinus* seedlings, PA-P = *Pinus arizonica*, PD-P = *P. durangensis*, PE-P = *P. engelmannii*, PL-P = *P. leiophylla* and PT-P = *P. teocote*. PA-H = hybrids of *Pinus arizonica* × *P. durangensis* genetically more similar to *P. arizonica*; PD-H = hybrids of *P. durangensis* × *P. arizonica* genetically more similar to *P. durangensis* and *P. durangensis* × *P. engelmannii* genetically more similar to *P. durangensis*; PE-H = hybrids of *P. engelmannii* × *P. arizonica* genetically more similar to *P. engelmannii*; PL-H = hybrids of *P. leiophylla* × *P. teocote* genetically more similar to *P. leiophylla*; PT-H = *P. leiophylla* × *P. teocote* genetically more similar to *P. teocote*; different letters indicate significant differences (α = 0.025).

**Table 4.** Descriptive statistics of the maximum height to the top of the needles (cm) of *Pinus* seedlings per species and their hybrids, in each trial and both trials together, after growing for 27 months in the field.

| Species | Mesa Alta | | | | Mesa Seca | | | | Overall, Both Trials | | | |
|---|---|---|---|---|---|---|---|---|---|---|---|---|
| | N | Median | Mean | Sd | N | Median | Mean | Sd | N | Median | Mean | Sd |
| PA-H | 5 | 31.4 b | 31.4 | 5.9 | 5 | 23.0 ab | 23.0 | 6.9 | 10 | 27.2 ab | 28.6 | 6.4 |
| PA-P | 3 | 31.6 b | 31.6 | 3.6 | 4 | 27.7 ab | 27.7 | 2.2 | 7 | 29.7 ab | 29.7 | 2.8 |
| PD-H | 18 | 44.5 a | 46.4 | 13.4 | 58 | 42.0 a | 43.9 | 12.9 | 76 | 42.0 a | 44.5 | 13.0 |
| PD-P | 14 | 42.1 a | 40.4 | 13.7 | 20 | 46.0 a | 47.7 | 14.2 | 34 | 42.9 a | 44.7 | 14.2 |
| PE-H | 56 | 32.8 b | 32.9 | 8.8 | 211 | 33.0 b | 34.3 | 8.5 | 267 | 33.0 b | 34.0 | 8.6 |
| PE-P | 78 | 32.0 b | 32.4 | 9.2 | 218 | 33.2 b | 34.6 | 9.6 | 296 | 33.0 b | 34.0 | 9.5 |
| PL-H | 7 | 34.0 a | 36.8 | 9.3 | 18 | 38.7 a | 39.3 | 10.3 | 25 | 38.4 ab | 38.6 | 9.9 |
| PL-P | 4 | 46.6 a | 44.7 | 16.1 | 13 | 42.0 a | 40.1 | 8.8 | 17 | 43.0 ab | 41.2 | 10.5 |
| PT-H | 49 | 40.2 a | 40.3 | 12.3 | 114 | 36.2 a | 37.4 | 10.4 | 163 | 36.0 a | 37.0 | 10.8 |
| PT-P | 24 | 39.6 a | 41.5 | 13.9 | 77 | 39.5 a | 42.1 | 16.7 | 101 | 35.1 a | 36.9 | 12.4 |
| H | 135 | 36.6 a | 37.5 | 9.9 | 406 | 34.6 a | 35.6 | 9.7 | 541 | 35.3 a | 36.5 | 9.7 |
| P | 123 | 38.4 a | 38.1 | 11.3 | 332 | 37.7 a | 38.4 | 10.4 | 455 | 36.7 a | 37.3 | 9.9 |

Sd = standard deviation, *N* = number of *Pinus* seedlings, PA-P = *Pinus arizonica*, PD-P = *P. durangensis*, PE-P = *P. engelmannii*, PL-P = *P. leiophylla* and PT-P = *P. teocote*. PA-H = hybrids of *Pinus arizonica* × *P. durangensis* genetically more similar to *P. arizonica*; PD-H = hybrids of *P. durangensis* × *P. arizonica* genetically more similar to *P. durangensis* and *P. durangensis* × *P. engelmannii* genetically more similar to *P. durangensis*; PE-H = hybrids of *P. engelmannii* × *P. arizonica* genetically more similar to *P. engelmannii*; PL-H = hybrids of *P. leiophylla* × *P. teocote* genetically more similar to *P. leiophylla*; PT-H = *P. leiophylla* × *P. teocote* genetically more similar to *P. teocote*; different letters indicate significant differences (α = 0.025).

## 2.3. Calculation of Stomatal Density in Needles

In both study sites, two needles per seedling were collected from 245 individuals (randomly chosen) of the three most frequent species (*P. engelmannii*, *P. durangensis*, and *P. teocote*) of the five initially considered species (as the required number of replicates was not obtained for the other two species). The needle samples were examined under a binocular stereoscope (EUROMEX: ED-1402-S) to calculate the stomatal density. The rows of stomata and the number of stomata within an area of one square millimeter were counted on both the abaxial and adaxial sides of the needles. The stomatal density of each face was calculated by multiplying the number of stomata per mm$^2$ by the number of rows. These values were summed to obtain the density per needle. We repeated this process with a second needle and then calculated the mean density. The values of central tendency and dispersion of the stomatal density were estimated (Table 4).

## 2.4. Statistical Analysis

Multiple median comparisons of the basal diameter, height at the apical bud, maximum height to the top of the needles and survival were made, and the respective *p* values were calculated for hybrid and pure species seedlings (Tables S1–S3). The comparisons were conducted using Nemenyi tests (posthoc.kruskal.nemenyi.test) and the PMCMR package of the R software version 1.4.1103 [24] (α = 0.025).

Spearman's analysis was used to examine the possible correlation ($r_s$) between the stomatal density and seedling diameter, height, and survival in both trials of hybrid and pure seedlings. Correlation values and their significance (*p*) were estimated considering α = 0.025.

The average survival (%) of the seedlings per seed stand was computed. Significant differences in the survival of the pure and hybrid seedlings were checked using the Delta index (*δ*) and the corresponding *p* value (Table S4). A *δ* value of zero indicates two collectives of individuals with identical survival rates, and a *δ* value of one indicates completely different survival rate (0 vs. 100% survival) [25,26]. The correlations between survival and stomatal density and the corresponding *p* values were also calculated.

## 3. Results

### 3.1. Growth Parameters and Survival

Considering both trials together and each trial separately, no significant difference between the basal diameter of the hybrid and pure seedlings was detected. Comparison of the diameter of the different species revealed that *Pinus engelmannii* seedlings were significantly larger than the *P. durangensis*, *P. leiophyilla,* and *P. teocote* seedlings (Table 2).

Considering both trials together and separately, no significant differences in the heights to the apical bud between hybrid and pure individuals of the corresponding species were observed. Overall, for both trials, the median height to the apical bud of the hybrid individuals was significantly larger than that of the pure seedlings (Table 3).

Considering both trials together and separately, comparison of the median values for the hybrid and pure species seedlings of *P. engelmannii* revealed significantly lower values of the maximum height to the top of the needles of this species relative to the hybrid and pure *P. durangensis* and *P. teocote* seedlings. On the other hand, there was no significant difference between hybrids and pure seedlings of each species in maximum height to the top of the needles (Table 4).

Considering both trials together, we observed significant differences in survival ($\delta$) between hybrid and pure species individuals of *P. arizonica* and *P. durangensis* (45% vs. 12% and 40% vs. 27%). However, we did not observe any significant differences in the analysis of the overall survival of hybrid and pure seedlings ($\delta = 0.23$, $p = 0.99$). Considering the trials separately, the mean survival rate of the *P. arizonica* hybrids was significantly higher than that of the pure individuals, but the mean survival of the hybrid *P. durangensis* was only higher in the Mesa Seca trial. Overall in both trials, pure *Pinus arizonica* seedlings exhibited the lowest survival in the field (12%), and the $\delta$ was significant relative to the other hybrids and pure seedlings. Pure *P. durangensis* seedlings presented significant $\delta$ relative to the hybrids and pure seedlings of *P. engelmannii* (26%, 46%, and 42%). Individuals of pure *P. teocote* (34%) and hybrids of *P. engelmannii* (46%) also presented significant $\delta$ (Table 5).

**Table 5.** Number of *Pinus* seedlings (*N*) growing in the field in both provenance trials, between July 2018 and October 2020 (classified as hybrids and pure species). Different letters in each survival column indicate significant differences in the survival rate.

| Species | Mesa Alta | | | Mesa Seca | | | Mean Survival (%) |
|---|---|---|---|---|---|---|---|
| | *N* 2018 | *N* 2020 | Survival (%) | *N* 2018 | *N* 2020 | Survival (%) | 2020 |
| PA-H | 11 | 5 | 45 a | 11 | 5 | 45 b | 45 ab |
| PA-P | 29 | 3 | 10 b | 29 | 4 | 14 d | 12 c |
| PD-H | 94 | 18 | 19 b | 94 | 58 | 61 ab | 40 a |
| PD-P | 66 | 14 | 21 b | 65 | 20 | 30 c | 26 b |
| PE-H | 294 | 56 | 19 b | 293 | 211 | 72 a | 46 a |
| PE-P | 354 | 78 | 22 b | 357 | 218 | 61 ab | 42 a |
| PL-H | 35 | 7 | 20 b | 34 | 18 | 52 b | 36 ab |
| PL-P | 27 | 4 | 15 b | 28 | 13 | 47 b | 31 ab |
| PT-H | 216 | 49 | 23 b | 215 | 114 | 52 b | 38 a |
| PT-P | 150 | 24 | 16 b | 150 | 77 | 51 b | 34 ab |
| H | 649 | 135 | 21 a | 649 | 406 | 63 a | 42 a |
| P | 628 | 123 | 20 a | 627 | 332 | 53 a | 36 a |

PA-P = *Pinus arizonica*, PD-P = *P. durangensis*, PE-P = *P. engelmannii*, PL-P = *P. leiophylla* and PT-P = *P. teocote*; PA-H = hybrids of *Pinus arizonica* × *Pinus durangensis* genetically more similar to *P. arizonica*; PD-H = hybrids of *P. durangensis* × *P. arizonica* genetically more similar to *P. durangensis* (13 live individuals) and *P. durangensis* × *P. engelmannii* genetically more similar to *P. durangensis* (63 live individuals); PE-H = hybrids of *P. engelmannii* × *P. arizonica* genetically more similar to *P. engelmannii*; PL-H = hybrids of *P. leiophylla* × *P. teocote* genetically more similar to *P. leiophylla*; PT-H = *P. leiophylla* × *P. teocote* genetically more similar to *P. teocote*; different letters indicate significant differences ($\alpha = 0.025$).

### 3.2. Stomatal Density and Growth in the Field

Considering both trials together, the stomatal density did not differ significantly between the seedlings of the same pure species and the respective hybrids. However, stomatal

density was significantly higher in the hybrid and pure individuals of *P. engelmannii* (145.7 in hybrids and 146.6 in pure individuals) than in the other two species ($p < 0.001$) (Table 6).

**Table 6.** Descriptive statistics of the estimated stomatal density (stomata/mm$^2$) for each pure species analyzed and the hybrids in both provenance trials.

| Species | $N$ Mesa Alta | $N$ Mesa Seca | Total $N$ | Median | Mean | Sd |
|---------|---------------|---------------|-----------|--------|------|-----|
| PD-H | 12 | 16 | 28 | 94.0 b | 94.6 | 17.1 |
| PD-P | 10 | 6 | 16 | 96.5 b | 97.7 | 24.6 |
| PE-H | 27 | 29 | 56 | 144.3 a | 145.7 | 33.6 |
| PE-P | 30 | 35 | 65 | 142.0 a | 146.6 | 35.6 |
| PT-H | 23 | 24 | 47 | 109.5 b | 111.3 | 24.2 |
| PT-P | 15 | 18 | 33 | 108.0 b | 112.2 | 30.3 |
| H | 62 | 69 | 131 | 119.7 a | 122.4 | 34.6 |
| P | 55 | 59 | 114 | 127.0 a | 129.6 | 38.0 |

Sd = standard deviation, $N$ = number of *Pinus* seedlings, PD-P = *Pinus durangensis*, PE-P = *P. engelmannii*, PT-P = *P. teocote*; PD-H = hybrids of *P. durangensis* × *P. arizonica* genetically more similar to *P. durangensis* and *P. durangensis* × *P. engelmannii* genetically more similar to *P. durangensis*; PE-H = hybrids of *P. engelmannii* × *P. arizonica* genetically more similar to *P. engelmannii*; PT-H = *P. leiophylla* × *P. teocote* genetically more similar to *P. teocote*. N = number of individuals; different letters indicate significant differences in the median stomata density ($\alpha = 0.025$).

In all three species, the mean stomatal density was significantly correlated with the basal diameter, height to the apical bud, maximum height to the top of the needles, and survival ($p < 0.01$). Stomatal density was significantly positively correlated with the basal diameter, but negatively correlated with both heights in the three species. In the hybrid seedlings, stomatal density was significantly positively correlated with survival at the seed stand level. In contrast, there was no significant correlation between these two variables in the pure seedlings. However, for the pure seedlings, a parabolic-like function with a negative quadratic coefficient of the survival rate with stomatal density was observed (with maximum survival rate at about 120 stomata/mm$^2$). These two variables were not significantly correlated in the pure seedlings. Consequently, very low and very high stomatal density correspond to lower survival of the pure seedlings (Figure 2 and Table 7). We did not detect a significant relationship between stomatal density and seedling growth or survival within any of the three species or their hybrids.

**Table 7.** Spearman's correlation ($r_s$) (and *p*-value) between the stomatal density and the basal diameter, height to the apical bud, and maximum height to the top of the needles per seedling. In addition, Spearman's correlations ($r_s$) (and *p*-value) between the stomatal density and survival in seedlings of the three selected pure pine species (*Pinus durangensis*, *P. engelmannii*, and *P. teocote*) and their hybrids (*P. durangensis* × *P. arizonica*, *P. durangensis* × *P. engelmannii*, *P. engelmannii* × *P. arizonica*, and *P. teocote* × *P. leiophylla*) ($\alpha = 0.025$) are shown for each trial.

| Variable | $r_s$ | *p*-Value |
|----------|-------|-----------|
| Basal diameter | +0.34 | $3 \times 10^{-7}$ |
| Height to apical bud | −0.36 | $5 \times 10^{-8}$ |
| Maximum height to the top of the needles | −0.23 | 0.0006 |
| Survival (hybrid seedlings) | +0.49 | $10^{-8}$ |
| Survival (pure seedlings) | −0.002 | 0.98 |

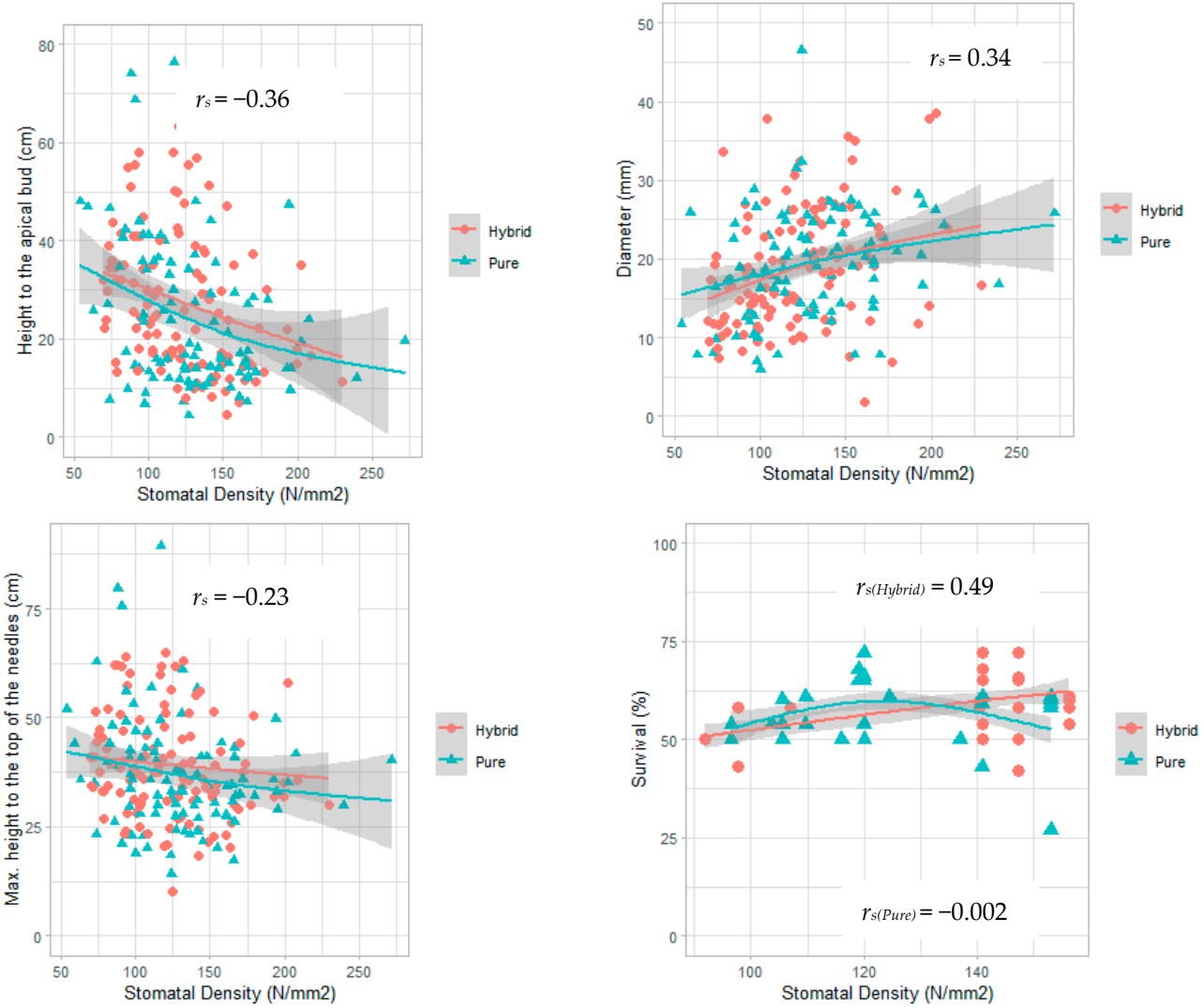

**Figure 2.** Correlations between stomatal density (N/mm$^2$) and basal diameter (mm), height to apical bud (cm), maximum height to the top of the needles (cm) per individual, and survival by seed stand in individuals of the three selected pure species (*Pinus durangensis*, *P. engelmannii*, and *P. teocote*) and their hybrids (*P. durangensis* × *P. arizonica*, *P. durangensis* × *P. engelmannii*, *P. engelmannii* × *P. arizonica*, and *P. tecote* × *P. leiophylla*). Mean values (cyan and red lines) and the 95% confidence level intervals for predictions (grey area) are based on generalized additive models (GAM).

## 4. Discussion

### 4.1. Seedling Growth and Survival

We observed significant differences in the growth of the height and basal diameter and a higher survival rate in *Pinus engelmannii* relative to the other species. These differences can be attributed to the fact that at the first stage of development, *P. engelmannii*, as a pioneer species, rapidly increases in diameter and needles, and to a lesser extent in height, and displays a cespitose growth habit [27,28]. The greater survival of this species is also due to its resistance to drought (mean annual precipitation from 670 to 830 mm [9]) and to the fact that it usually grows on plateaus, slopes, valleys, and terraces, at elevations of between 1500 and 2700 m [29,30]. The same applies to the more drought-resistant species of *P. leiophylla* and *P. teocote* [9], which are often associated with *P. engelmannii* [8]. Although *P. arizonica* is one of the most important timber species in the SMO [31], survival of these seedlings was lower than that of other species in both trials (Table 5). This species has specific growth requirements, including a pH of 4.9 ± 0.3 [32], dense tree cover [33,34], and

a mean annual precipitation of between 870 and 1200 mm [9]. These conditions did not occur in the study area, as tree cover is sparse in both sites, and the level of precipitation was low in the year prior to data collection.

In general, the growth of hybrid seedlings was similar to that of the pure seedlings in the trials (Tables 2–4). Only the overall median height to the apical bud of hybrid seedlings was slightly, but significantly, greater than the median height of the pure seedlings (Table 3). This has also been observed in hybrids of other species, such as *Pinus oocarpa* × *P. pringei* [19] and *P. arizonica* × *P. engelmannii* [29], and adult hybrid trees of *Pinus luzmariae* × *P. herrerae*, which were taller than pure *P. luzmariae* trees [35]. However, hybrids generally show intermediate values of height and diameter relative to the (pure) parents [36].

In addition, the survival of hybrids of *P. arizonica* and of *P. durangensis* was significantly higher than that of the pure species (Table 5), which may indicate hybrid vigor (heterosis). In theory, heterosis occurs for different reasons: heterozygous individuals display higher levels of fitness than homozygous individuals, thus favoring the survival of hybrids [37]. Individuals with higher individual heterozygosity show more stable growth, being less affected by environmental factors [20], because hybrids have greater genetic variability, which allows them to adapt to a greater number of ecosystems and conditions [5].

Another reason for hybrid vigor is overdominance, which occurs when heterozygous individuals are more vigorous than homozygous individuals, giving rise to superior hybrids [37,38]. Dominance occurs when less homozygous individuals have, by definition, lower values of inbreeding and lower inbreeding depression [39–41].

However, the most likely cause of hybrid vigor is that the environmental conditions, in both trials, were more suitable for the hybrids with *P. engelmannii* genes (*P. arizonica* × *P. engelmannii*, *P. durangensis* × *P. engelmannii*) (Table 5).

Finally, after growing for 27 months in the field, the mean survival rate of seedlings was 35% (Table 2). This is lower than the rate determined by Mejía et al. [42], who studied seedlings of *Pinus* of different species and ages in the SMO, where only plantings trees older than eight years had a survival rate of less than 60%. This is also lower than the rate reported by Benítez [43], who calculated a mean survival of 60% in *Pinus engelmannii* plantations in Durango, Mexico. However, it was similar to that in the plantations studied by Torres Rojo [44], with a mean survival rate of 38%, and to some of the *Pinus engelmannii* plantations studied by Prieto Ruíz et al. [45]. Possible causes of low survival in the planting sites (both trials) include the presence of *Pappogeomys castanops* Baird and low rainfall during 2019 (429 mm), the year prior to data collection [46].

*4.2. Stomatal Density*

Stomatal density was the highest in the drought-tolerant *P. engelmannii* [26,30] (Table 7). This finding is consistent with those of other studies that have reported a higher stomatal density and/or number of stomatal rows in *P. ponderosa* [47] and in some Mediterranean pines under drought conditions [48–50]. According to Afas et al. [51] and Shu [47], a higher stomatal density could enable increased leaf gas exchange during short, favorable periods and greater control of water loss and gas exchange under drought stress in harsh dry conditions. Stomatal density depends on different environmental factors, such as water stress [52] and changes in ambient $CO_2$ concentration [53]. Despite the influence of environmental factors, stomatal density is strongly controlled by genetic factors [54].

The stomatal density was positively and significantly correlated with the basal diameter and negatively correlated with height in the pine species analyzed (Table 7, Figure 2). In a study of an $F_1$ hybrid between *Quercus robur* and *Q. robur* subsp. *Slavonica*, Gailing et al. [55] also observed a positive, significant correlation between the stomatal density and basal diameter; however, the correlation between stomatal density and height was also positive. We obtained the opposite result regarding height, which can be explained by the lower height growth of *P. engelmannii* than of *P. durangensis* and *P. teocote* (Tables 3 and 4). However, *P. engelmannii* has the highest stomatal density (Table 6). The same authors [55] also stated that (i) stomatal development is regulated by different genetic and environmen-

tal signals and (ii) in *Q. robur*, the allele of a particular quantitative trait locus associated with higher stomatal density was generally correlated with taller plants and an increase in size, indicating pleiotropic gene effects or a close genetic linkage, as also reported by Chebib and Guillaume [56].

In the present study, in addition to influencing growth, stomatal density was also positively correlated with the survival of the hybrid seedlings. However, for the pure seedlings, plotting the survival rate against the stomatal density yielded a parabolic-like curve (Figure 2 and Table 7). Thus, on average, hybrids with a high stomatal density survived better than pure seedlings with a similar high stomatal density, which could be explained by differences in other traits not studied here. These other traits could have contributed to enhancing the adaptive capacity of the hybrids by enabling them to cope with environmental conditions in both trials more successfully than the pure individuals. Significant variation in stomatal density has been detected between clones and hybrids of *Populus* species, and its respective correlation with biomass production [57] and light conductance [58], indicating that stomatal density may vary among clones or pure species, as well as among hybrids; such variation will enable the trees to adapt to the surrounding environmental conditions.

## 5. Conclusions

The basal diameter, height to the apical bud, and maximum height to the top of the needles varied weakly between hybrid and pure seedlings of different pine species. A greater height to the apical bud and survival of hybrids were detected in *Pinus arizonica* and *P. durangensis* than in the pure species. After growing for 27 months in the field, the hybrids generally displayed the same capacity as the pure seedlings (and in some cases a greater capacity) to withstand weather conditions, survive, and grow effectively. These differences are expected to increase over time in the field. Thus, there is no reason to exclude these hybrids from the forest management plans.

We recommend continuing to monitor these trials in order to determine the long-term viability of the hybrid and pure seedlings. Because of the spatial and temporal limitations of the study, we also suggest replicating this type of trial with other species and in other sites, as there is a wide variation in the chances of detecting hybrid vigor.

The results of this research will help forest managers to select the most appropriate species and their hybrids for reforestation or plantations, thus contributing to sustainable forest protection, conservation, and management, including adaptive silviculture, and to satisfying the growing demand for wood in the forestry sector.

**Supplementary Materials:** The following supporting information can be downloaded at: https://www.mdpi.com/article/10.3390/f13111791/s1, Table S1: Difference between basal diameters medians of *Pinus* species in millimeters and *p*-values calculated in Kruskal-Wallis multiple testing (both trials); $\alpha = 0.025$; Table S2: Differences between the median height to the apical bud of *Pinus* species in centimeters, and the corresponding *p*-values, calculated in Kruskal-Wallis multiple tests (overall for both trials); $\alpha = 0.025$; Table S3: Difference between the medians of the maximum height to the top of the needles of *Pinus* species, in centimeters, and *p*-values calculated in Kruskal-Wallis multiple tests (overall for both trials); $\alpha = 0.025$; Table S4: Delta index ($\delta$) and corresponding *p*-values in delta tests for the survival of *Pinus* species (both trials combined), $\alpha = 0.025$.

**Author Contributions:** Conceptualization, C.W.; formal analysis, R.S.S.-H.; investigation, R.S.S.-H.; methodology; supervision, C.W.; writing of the original draft, R.S.S.-H. and C.W.; writing—reviewing and editing, C.W., C.Z.Q.-P., J.C.H.-D. and J.Á.P.-R. All authors have read and agreed to the published version of the manuscript.

**Funding:** This study was supported by the Council of Science and Technology of the state of Durango (COCYTED); Finance Code 21571/2020. We would like to thank the Science and Technology Council (CONACYT) for a postgraduate scholarship, which acted as an incentive to carry out the study. The funding bodies had no role in the study design, data collection and analysis, decision to publish, or preparation of the manuscript.

**Acknowledgments:** We are thankful to the administration of the ejidos Ciénega de Salpica el Agua y Laguna de la Chaparra, municipality of Santiago Papasquiaro, state of Durango, México (1005) (Engineer Fernando Salazar Jiménez) for their helpful assistance.

**Conflicts of Interest:** The authors declare no conflict of interest.

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
