# Peer review of "Field Trials to Assess the Growth, Survival, and Stomatal Densities of Five Mexican Pine Species and Their Hybrids under Common Plantation Conditions"

_forests, doi:10.3390/f13111791_

Round 1
Reviewer 1 Report
The manuscript presents a series of tables and statistically differences between the growth of hybrid and pure species seedlings. Results should also highlight the influence of field trials.

Author Response
Greetings and thanks for your corrections, in the text all the changed data is marked in yellow.
Response to reviewer 1.
Field trials to assess growth and stomatal densities of five Mexican pine species and their hybrids,
under plantation conditions.
In this manuscript, the authors propose to evaluate and compare the growth of seedlings from five
species of pine, pure species and hybrids and stomatal densities for only three of these species.
The structure and the presentation of the manuscript have to be improved.
The entire reference list must be check. All the references in the text have to be check. There are
55 references mentioned in the text and 57 in the reference list.
Response: We have checked and modified the number and order of the 57 references, so they
match.
See, for example, L.48. Two field trials were established, one in La Mesa Alta and another in La
Mesa Seca. But, all the results are provided for both sites together. Why did you not provide
information about seedlings separately for each trial? In this form, the paper should not be
published in Forests.
Response: The information about seedlings was separated for each trial and was included in
Tables 2-4.
L 21. 534 hybrid and 450 pure species seedlings, see TABLE 2 (L 128-130). Please check the
number of seedlings.
Response: That table is now Table 5, and the numbers were corrected, there were some
mismatches.
L24-25. The mean height of the apical bud was presented in Table 3b. No information about
differences between the mean height of the apical bud was provided. See table 6 where the
median height of the apical bud was analyzed (L203-205).
Response: That information is now available in Table 3.
Table 6 was sent to the supplementary information.
L90. Seedlings not trees
Response: corrected see line 90
L 106. Please provide information about the seedlings, at least age.
Response: information such as age of the seedlings is available in Materials and Methods (lines 96-
97).
L128-135. Table 2 should be moved to section 3. Results. There are 737 seedlings in Mesa Seca
(2020) not 738. Please check and change
Response: Table 2 was moved and it is now Table 5. The correct number of seedlings in Mesa Seca
is 738.
L137. 454 pure seedlings, not 455. Please check and change
Response: The numbers inside the table were corrected, 455 was the right number.
L146-152. Table 3a, Table 3b and Table 3c should be moved to section 3. Results
Response: The Tables were moved to section 3, and the numeration changed to 2, 3 and 4.
L154. Why did you consider only 3 tree species, not all 5 species analyzed?
Response: the other two didn’t reach the required number of repetitions. Lines 133-134
L164-167. Table 4 should be moved to section 3. Results
Response: The Table was moved to section 3 and now is Table 6.
L186-196 Significantly differences were recorded for PE and PD-H, PE and PT. Please check and
change
Response: We wrote (lines 162-164): “Comparison of the diameter of the different species
revealed that Pinus engelmannii seedlings were significantly larger than seedlings of P.
durangensis, P. leiophyilla and P. teocote (Table 2)”.
L 199-201. Why did you highlight only the differences between hybrid and pure seedlings all
over the sites and species? There are significant differences between PE-H and PD-H for
example. See Table 6 (L203-209).
Response: All the significant values were highlighted, see supplementary information.
L213-214. Please, check Table 7 (differences between PD and PT).
Response: We don’t see any mistake; the p-values are not significant between PD and PT., the
table is now in Table S3.
Please change L227-229. Where are the survival rates considering the two trials separately?
Response: We wrote (lines 218-220): “Also separating the two trials, the P. arizonica hybrid´s
mean had a significantly higher survival rate than pure species individuals, but the P. durangensis
hybrid’s mean was larger only in the Mesa Seca trial.” The survival rates per site are available in
Table 5.
L256 Please provide the correlation value on the graphs (figure 2)
Response: The correction values were added to the graphs
L300-301. See PA, PD, PL and PT.
Response: Growth of hybrid seedlings in the trials was different to that of the pure seedlings of the
species under study. This has not been observed in hybrids of other species.
Please check and correct L340.
Response: We did not find any mistake in that line.
There are no conclusions about the stomatal density.
Response: we wrote (lines 345-346): “Stomatal density has a clear influence in growth of pine
seedlings, the effect of stomatal density on survival, varied between hybrids and pure species
individuals”.
Reviewer 2 Report
Manuscript Number: forests-1908825
Title: Field trials to assess growth and stomatal densities of five Mexican pine species and their hybrids, under plantation conditions
Authors: Sánchez-Hernández RS, et al.
Remarks:
This submitted paper examined interspecific differences and the hybrid preferences in survival, growth and stomatal traits of Mexican pine species based on the common-plantation trials. The authors found that some growth traits and stomatal density of the hybrids were greater than the pure species, and that the stomatal density was positively correlated with the growth performances. I think that insights obtained in the study are important for ongoing forestation, forest breeding and management programs of the pine species.
However, the present version of the paper has some major problems, mainly on the insufficient explanation of backgrounds (Introduction) and interpretations/indications (Discussion), and the unclear presentation of methodology and results. The paper also has unnecessary sentences with little contribution to the paper and many inappropriate and unclear word uses throughout the manuscript, which should be refined. So I recommend the authors to reconsider carefully the paper and resubmit, according to the major and minor suggestion pointed as below.
Major Comments:
1. The Introduction section has too much paragraphs; which should be reconstructed to 4-5 paragraphs, according to the revised construction of the main theme of the study.
The authors wrote in Lines 87- “The present research aims, for the first time, … seedlings planted in the field”. But, one major problem is that there are no logical statements on what is the first time the authors investigate and why the authors investigate it. The authors should rewrite, in front of here, what has been studied but what is unknown and necessary to be clarified, together with the relevant backgrounds.
And, the Introduction has many necessary sentences of explaining the individual studies (e.g., Lines 48-52), the study materials (e.g., Lines 55-57) and the results of the study (e.g., Lines 94-98), which makes the paper verbose and unsystematic. These explanations should be written in other sections (Methods, Results and Discussion sections). The authors should check and reconsider throughout this section once again and move the sentences into the appropriate positions.
2. In Results, presentations of Tables 5-8, on the differences in growth and stomatal traits among species and between mating groups (hybrids or pure crosses), are rather difficult to be seen and also of too volume, and therefore not effectively informative. I recommend strongly the authors to cut these tables and include the statistical significances of multiple differences into each of the former tables (Tables 2-4) using different alphabets (a, ab, b, and so on).
3. In the Discussion section, the authors merely say the correspondences of the results with previous studies (as in Lines 328-334) or the reasons of results based on the data structure (as in Lines 335-339). There are no discussions on the mechanical and fundamental reasons of the obtained results. I suggest the authors to reconsider and interpret the results, on why the results are brought and what the results indicate for the main theme of the study. Otherwise the readers cannot understand how the results contribute to the resolution of the main theme of the study and cannot appreciate the value of the study. Please refine this section.
Minor Comments:
Title
4. “under plantation conditions” should be “under commom-plantation conditions”.
Abstract
5. Lines 16-. Regarding also with the Major comment #1, the first and second sentences are not effectively connected to the third and later sentences, because there is no explanation on the significance of the study. Please reconsider to rewrite here why the authors studied.
6. Lines 28-. Regarding also with the Major comment #3, it is better to write what the results indicate and what the authors suggest for the main theme of the study.
Introduction
7. Lines 35-. “more new” should be like “more different” or “more diverse”. Check also the authors’ intended meaning.
8. Lines 68-71. This paragraph (about adaptive strategy of morphology) is too general and also different from the scope of the present study (hybrid preference). I think that it is unnecessary.
Materials and Methods
9. Lines 126-. Although wrote in the tables, when the seedling survival and growth traits were measured should be explained also in the main text.
10. Table 2. Please correct the bold and the underline in P. arizonica hybrid.
11. Lines 137-. It is unclear where the numbers of measured trees (541 + 455 = 996) were come from. There should be an explanation that the number was trees which survived in October 2020 (258 + 738 in Table 2).
12. Tables 3a-c. Regarding also with the Major comment #2, on revising these tables, please also add the sample numbers of each species (as in Table 4).
13. Lines 154- and Table 4. I don’t catch where the number of samples on the stomatal density measurements for each species were come from. Also, it is unclear what numbers of trees were measured in each of two sites. If the authors selected trees for the measurement randomly or based on any criteria, please explain so.
Results
14. Lines 185-. The heading of subsection 3.1 “Analysis of growth parameters …” should be “Growth parameters …”.
15. Lines 189-. “thicker” should be like “larger”.
Discussion
16. Lines 266-. The headings of subsections were not uniform (4.1, 4.1.2, 4.1.3, …). The italic is not also uniform. Please correct.
17. Lines 268-, and so on. I don’t catch the words “under study” used throughout this section. Please reconsider.
18. Lines 276-. “lesser” should be “less”.
19. Lines 279-. “In the present study” here and “In this study” (as in Lines 321-, and so on) should be uniform throughout the manuscript.
Author Response
Greetings and thanks for your corrections, in the text all the changed data is marked in yellow.
Response to reviewer 2:
However, the present version of the paper has some major problems, mainly on the insufficient
explanation of backgrounds (Introduction) and interpretations/indications (Discussion), and the
unclear presentation of methodology and results. The paper also has unnecessary sentences with
little contribution to the paper and many inappropriate and unclear word uses throughout the
manuscript, which should be refined. So I recommend the authors to reconsider carefully the paper
and resubmit, according to the major and minor suggestion pointed as below.
Major Comments:
1. The Introduction section has too much paragraphs; which should be reconstructed to
4-5 paragraphs, according to the revised construction of the main theme of the study.
Response: We reconstructed the introduction as suggested.
The authors wrote in Lines 87- “The present research aims, for the first time, … seedlings
planted in the field”. But, one major problem is that there are no logical statements on what
is the first time the authors investigate and why the authors investigate it. The authors
should rewrite, in front of here, what has been studied but what is unknown and necessary
to be clarified, together with the relevant backgrounds.
Response: We reworded the paragraph to (Lines 76-81): “The present research aims to evaluate
and compare the growth and survival rate between putative hybrid seedlings and seedlings of the
pure parental species of Pinus arizonica, P. durangensis, P. engelmannii, P. leiophylla and P.
teocote, in two reciprocal trials in the Sierra Madre Occidental (SMO) of the state of Durango,
México. Our aims also included examining the possible correlation between the stomatal density
of the needles and the growth and survival of the seedlings.”
And, the Introduction has many unnecessary sentences of explaining the individual studies
(e.g., Lines 48-52), the study materials (e.g., Lines 55-57) and the results of the study (e.g.,
Lines 94-98), which makes the paper verbose and unsystematic. These explanations should
be written in other sections (Methods, Results and Discussion sections). The authors should
check and reconsider throughout this section once again and move the sentences into the
appropriate positions.
Response: We reconstructed the introduction and distributed the information in different sections.
. In Results, presentations of Tables 5-8, on the differences in growth and stomatal traits
among species and between mating groups (hybrids or pure crosses), are rather difficult to
be seen and also a lot of volume, and therefore not effectively informative. I recommend
strongly the authors to cut these tables and include the statistical significances of multiple
differences into each of the former tables (Tables 2-4) using different alphabets (a, ab, b, and
so on).
Response: These tables were sent to Appendixes, and the alphabets (a, ab, b, and so on) were used.
3. In the Discussion section, the authors merely say the correspondences of the results with
previous studies (as in Lines 328-334) or the reasons of results based on the data structure
(as in Lines 335-339). There are no discussions on the mechanical and fundamental reasons
of the obtained results. I suggest the authors to reconsider and interpret the results, on why
the results are brought and what the results indicate for the main theme of the study.
Otherwise the readers cannot understand how the results contribute to the resolution of the
main theme of the study and cannot appreciate the value of the study. Please refine this
section.
Response: We improved the discussion section with new information, see lines 290-298 and 325-331.
Minor Comments:
Title
4. “under plantation conditions” should be “under commom-plantation conditions”.
Response: We modified the title, as the recommendation.
Abstract
5. Lines 16-. Regarding also with the Major comment #1, the first and second sentences
are not effectively connected to the third and later sentences, because there is no
explanation on the significance of the study. Please reconsider to rewrite here why the
authors studied.
Response: We wrote (Lines 15-24): “Understanding hybridization is important for practical
reasons, as the presence of hybrids in seed stands can influence the success of natural regeneration
and reforestations. Hybridization creates new gene combinations, which can promote or enhance
adaptation to new or changing environments.”
6. Lines 28-. Regarding also with the Major comment #3, it is better to write what the
results indicate and what the authors suggest for the main theme of the study.
We eliminated the method part and included (lines: 29-31): “In summary, Pinus hybrids generally
had the same and, for some species, greater ability than pure species to withstand weather
conditions, survive, and grow effectively in the two growth trials.”
Introduction
7. Lines 35-. “more new” should be like “more different” or “more diverse”. Check also
the authors’ intended meaning.
Response: We wrote (line 39): “more different genetic material than the generated directly by
mutations [3].”
8. Lines 68-71. This paragraph (about adaptive strategy of morphology) is too general and
also different from the scope of the present study (hybrid preference). I think that it is
unnecessary.
Response: The paragraph was deleted.
Materials and Methods
9. Lines 126-. Although wrote in the tables, when the seedling survival and growth traits
were measured should be explained also in the main text.
Response: We wrote (lines 116-118): “In October 2020, seedling survival was calculated per pure
species and hybrid and per trial, as a percentage of the total number of individuals that were
planted in both trials in July 2018 (Table 2).”
10. Table 2. Please correct the bold and the underline in P. arizonica hybrid.
Response: The table format was corrected, now is Table 5.
11. Lines 137-. It is unclear where the numbers of measured trees (541 + 455 = 996) were come
from. There should be an explanation that the number was trees which survived in October
2020 (258 + 738 in Table 2).
Response: We wrote (Table 2): “In total, 541 hybrid seedlings and 455 pure species seedlings of
the five species studied were analyzed (258 from Mesa Alta and 738 from Mesa Seca)”.
12. Tables 3a-c. Regarding also with the Major comment #2, on revising these tables, please
also add the sample numbers of each species (as in Table 4).
Response: We added the number of individuals of each species to the tables
13. Lines 154- and Table 4. I don’t catch where the number of samples on the stomatal density
measurements for each species were come from. Also, it is unclear what numbers of trees
were measured in each of two sites. If the authors selected trees for the measurement
randomly or based on any criteria, please explain so.
Response: We added the number of individuals of each site to Table 6 (before it was Table 4).
Results
14. Lines 185-. The heading of subsection 3.1 “Analysis of growth parameters …” should be
“Growth parameters …”.
Response: we changed the title of the table to (LINE 159): “3.1 Growth Parameters and Survival”.
15. Lines 189-. “thicker” should be like “larger”.
Response: we wrote (Line 163): “larger than seedlings of P. durangensis, P. leiophyilla and P.
teocote (Table 5).”
Discussion
16. Lines 266-. The headings of subsections were not uniform (4.1, 4.1.2, 4.1.3, …). The italic is
not also uniform. Please correct.
Response: we shortened the subsections of the discussion section and uniformed the format.
17. Lines 268-, and so on. I don’t catch the words “under study” used throughout this section.
Please reconsider.
Response: The “under study” phrases were deleted or changed for “analyzed species”.
18. Lines 276-. “lesser” should be “less”.
We wrote: and to a less extent in height, and shows cespitose growth. [33, 34]. Line 279.
19. Lines 279-. “In the present study” here and “In this study” (as in Lines 321-, and so on)
should be uniform throughout the manuscript.
Response: The “In this study” phrases were deleted or changed for “In the present study”
Reviewer 3 Report
The manuscript is well-written, and the data clearly presented and well-organized, and illustrated. Following are some suggestions and questions and minor editorials:
Line 2-3: Please use uppercase for each word in the title, subtitles, and subheadings
Line 64: geographic, and
Line 65: allow the measurement
Line 75: concentration, and
Line 99-100: 2. Materials and Methods 2.1. Study Site
Line 102: you may delete “as”
Line 107: evaluated, and
Line 117: The topography of the area
Line 121: that consume
Line 126: Please double-check if this is per seedling stand or seed stand?
Line 142: is considered
Line 248: than in the other
Line 252: basal diameter but
Line 345: a greater capacity
Line 339: The discussion needs to include the limitations of the current study and further experiments to answer important questions resulting from this study. Are there any implications for future research?

Author Response
Response to reviewer #3.
The manuscript is well-written, and the data clearly presented and well-organized, and illustrated.
Following are some suggestions and questions and minor editorials:
Line 2-3: Please use uppercase for each word in the title, subtitles, and subheadings
Response: We put uppercase in each main word in the title, subtitles, and subheadings.
Line 64: geographic, and climatic factors
Response: We added the comma to the sentence
Line 65: allow the measurement
Response: We wrote (lines 59-60): “only allow measurement of phenotypic differences.”
Line 75: concentration, and
Response: We added the comma to the sentence, line 69.
Line 99-100: 2. Materials and Methods 2.1. Study Site
Response: We changed the numeration (Lines 86-87).
“2. Materials and Methods
2.1. Study Site.”
Line 102: you may delete “as”
Response: We wrote (lines 89-90): “named also hybrid seedlings and pure species seedlings”.
Line 107: evaluated, and
Response: We added the comma to the sentence, line 94.
Line 117: The topography of the area
Response: We wrote (line 107): “The topography of the area consists…”.
Line 121: that consume
Response: We wrote (line 111): “species that consume plants or parts of plants”.
Line 126: Please double-check if this is per seedling stand or seed stand?
Response: We wrote (Table 2): “In October 2020, seedling survival was calculated per pure species
and hybrid and per trial, as a percentage of the total number of individuals that were planted in
both trials in July 2018”.
Line 142: is considered
Response: We wrote (line 127): “Needle length in young seedlings is considered a good indicator
of future growth”.
Line 248: than in the other
Response: We wrote (line 242): “than in the other two pine species”
Line 252: basal diameter but
Response: We wrote (lines 261-262): “basal diameter, but negatively correlated…”
Line 345: a greater capacity
Response: We wrote (line 360): “in some species, a greater capacity than pure species…”.
Line 339: The discussion needs to include the limitations of the current study and further
experiments to answer important questions resulting from this study. Are there any
implications for future research?
Response: We wrote (lines 365-366): “Because of the study’s limits in space and scale, we also
suggest to replicate this type of trial with other species and sites, as there is a wide variation in the
chances of detecting hybrid vigour. “
(line 368): “The results of this research will help forest managers to make better decisions when
selecting species and their hybrids for reforestation or plantations, thus contributing to forest
protection and conservation and to satisfy the growing demand for wood in the forestry sector.”.
Round 2
Reviewer 1 Report
Thank you for considering the comments and suggestions I made.
Author Response
We thanked the reviser for the time and effort dedicated to this manuscript.
Reviewer 2 Report
Manuscript Number: forests-1908825-v2
Title: Field trials to assess growth, survival and stomatal densities of five Mexican pine species and their hybrids, under common-plantation conditions
Authors: Sánchez-Hernández RS, et al.
Remarks:
The revised version of the paper has improved substantially. It becomes more readable by the readers, especially regarding with the clearer backgrounds of the study in Introduction. I consider that the paper is closer to be acceptable for this journal. I comment on some points which the authors should reconsider once again the table presentation, the discussion and the word uses.
Major and Minor Comments:
Abstract
1. Lines 21-. “Our aims also included examining …” could be “We also examined …”.
2. Lines 31-. “measures” could be “options” (Does this reflect the authors’ meaning?).
Results
3. Table 2. I wonder that basal diameters are around 10-20 cm in 27 months after planted. Please check a unit.
4. Tables 3 and 4. Please show the units of the heights.
5. Tables 5 and 6. Don’t the authors show the total values and differences of hybrids and pure species (“H” and “P”), as shown in the Tables 2-4?
6. Table 6. Why the authors show mean(sd), maximum and minimum values here, different from the median and mean(sd) values as shown in Tables 2-4? I think that the presentations could be uniform over tables.
7. Table 6. Only here the order of alphabets is different from Tables 2-5. Alphabets (a, b, …) should be placed in the order of significantly larger group.
Discussion
8. Lines 325-. Please write the authors’ names (as “According to xxx et al. [51] and xxx et al. [47]”) at this occasion.
9. Lines 333-339. In this paragraph, as suggested in the time of draft-review (Major comment #3), the authors merely say the correspondences of the results with previous studies, and there are no interpretations why the stomatal density was differently correlated with diameter and heights. I suggest once again the authors to reconsider what the results indicate mechanically and fundamentally for the main theme of the study (hybrid vigor).
10. Lines 341-349. In this paragraph also, as suggested in the time of draft-review (Major comment #3), the authors merely say the correspondences of the results with previous studies, and there are no interpretations why the relationship between stomatal density and survival was different between hybrids and the pure species (i.e., why the survival of the pure species is high at the medium stomatal density). I suggest once again the authors to reconsider what the results indicate mechanically and fundamentally for the main theme of the study (hybrid vigor).
